# Public acceptability of the UK Soft Drinks Industry Levy: repeat cross-sectional analysis of the International Food Policy Study (2017–2019)

Jean Adams ,[1] David Pell,[1,2] Tarra L Penney,[3] David Hammond,[4] Lana Vanderlee,[5] Martin White[1]

¹MRC Epidemiology Unit, University of Cambridge, Cambridge, UK
²Midlands Region, NHS RightCare, Newcastle upon Tyne, UK
³School of Global Health, York University Faculty of Health, Toronto, Ontario, Canada
⁴School of Public Health, University of Waterloo, Waterloo, Ontario, Canada
⁵School of Nutrition, Laval University, Quebec, Quebec, Canada

**Correspondence to**
Dr Jean Adams;
jma79@medschl.cam.ac.uk

## ABSTRACT

**Objectives** To determine whether public acceptability, in terms of both support for and perceived effectiveness of, the UK Soft Drinks Industry Levy (SDIL) changed between 4 months prior to, and 8 and 20 months after, implementation.

**Design** Repeat cross-sectional online survey.

**Setting** The UK.

**Participants** UK respondents to the International Food Policy Study aged 18–64 years who provided information on all variables of interest in November–December 2017 (4 months prior to SDIL implementation), 2018 (8 months after) or 2019 (20 months after; n=10 284).

**Outcome measures** Self-reported support for, and perceived effectiveness of, the SDIL.

**Results** The adjusted logistic regression model predicted that 70% (95% CI: 68% to 72%) of participants supported the SDIL in 2017, 68% (95% CI: 67% to 70%) in 2018 and 68% (95% CI: 66% to 70%) in 2019. There was no evidence of a difference in support in 2018 vs 2017 (OR: 0.93; 95% CI: 0.81 to 1.05); or in 2019 vs 2017 (OR: 0.90; 95% CI: 0.78 to 1.03). The adjusted logistic regression model predicted that 72% (95% CI: 70% to 74%) of participants perceived the SDIL to be effective in 2017, 67% (95% CI: 65% to 69%) in 2018 and 67% (95% CI: 64% to 69%) in 2019. There was evidence that perceived effectiveness decreased a small amount in 2018 vs 2017 (OR: 0.78; 95% CI: 0.69 to 0.88). The difference in 2019 vs 2017 was similar.

**Conclusions** We found high support for the SDIL among UK adults and this did not change between 4 months before implementation and 8 or 20 months after. While perceived effectiveness remained high, there was evidence that this decreased slightly after implementation in 2018, but no further in 2019. Greater understanding of influences on public acceptability of effective structural public health interventions is required.

## INTRODUCTION

Taxes on sugar-sweetened beverages (SSBs) are recommended by the WHO to reduce sugar consumption and prevent non-communicable diseases.[1] Systematic review evidence suggests that SSB taxes lead to reductions in SSB purchasing and consumption,

### Strengths and limitations of this study

► We used three annual waves of a large, population-based survey (n=10 284).

► We were careful to present the Soft Drinks Industry Levy (SDIL) as an intervention targeted at manufacturers rather than consumers, with revenues earmarked for health-promotion activities.

► This is a repeat cross-sectional design with measures before and after implementation of the SDIL in April 2018, but all time points were after announcement of the SDIL in March 2016.

► While all measures have strong face validity, we have not explored other aspects of validity or reliability of any of the measures used; in many cases it would be hard to know what the 'gold standard' measure should be.

but there is substantial heterogeneity in effect sizes[2] and tax design.[3] Better understanding of the contextual factors that influence the effectiveness of SSB taxes may enable taxes to be better tailored to context.[4] Such contextual factors may include public acceptability of SSB taxes.

By their nature, regulatory policies such as SSB taxes require political support for implementation. Political support is, in turn, likely to be influenced by public acceptability. As well as influencing implementation, public acceptability may also influence the effectiveness and longevity of SSB taxes.[5] For example, if price increases following an SSB tax are not acceptable to the public, then they may travel to buy SSBs in untaxed areas (so-called cross-border shopping).[3] Further, a number of food taxes have been repealed after implementation, in part due to perceived public backlash.[6–8] This makes it important to understand how public acceptability of SSB taxes changes after implementation.

Public acceptability of a policy reflects both public support for that policy and perceptions of how effective the policy may be.[9 10] Public support for hypothetical SSB taxes ranges from around 35% to 60%.[11–28] A recent systematic review reported a pooled figure for support of 42% (95% CI: 38% to 47%).[9] Associations between support and variables such as age, sex, SSB consumption and socioeconomic position are inconsistent.[11 13 16 18 24–26 28–30] However, support is consistently higher when it is clear that revenue raised will be used for health-promotion activities, such as subsidies on healthy food.[9 19 20 27 30 31] There is also evidence that framing of SSB taxes, particularly how SSBs are defined and what the stated aims of taxes are, can influence public support.[32]

A recent systematic review of randomised controlled trials indicates that providing information on the effectiveness (or ineffectiveness) of government policies leads to significant changes in support for those policies.[10] Perceived effectiveness of SSB taxes has been less studied than public support.[9 18 30] However, in a systematic review, pooled estimates were that 39% (95% CI: 26% to 54%) of the public believe SSB taxes reduce purchases and consumption, and 40% (95% CI: 29% to 54%) believe that they impact on health-related outcomes.[9] A perception that SSB taxes are unlikely to be effective is a common explanation for low public support in qualitative studies.[9 17 19]

Most previous work on public acceptability of SSB taxes has focused on hypothetical taxes. A systematic review of support for government interventions to change health-related behaviours (that did not include any studies on SSB taxes) found that support tends to be higher for implemented, rather than hypothetical, policies.[33] Similarly, perceived effectiveness may be influenced by whether respondents are reporting on a hypothetical or implemented tax.

We are aware of four studies on public acceptability of implemented SSB taxes. One study of the French excise tax on sweetened beverages found 49% of the public supported the tax and 58% believed it would improve health.[30] An international study exploring public support for a range of dietary public health policies in 2017 included data from Mexico, where an SSB tax was implemented in 2014. While support for an SSB tax was higher in Mexico than other countries (54% vs 30%–49%), the same was true for many other policies studied.[28] Two studies have focused on the UK Soft Drinks Industry Levy (SDIL; described in box 1).[34 35] Our previous population-based survey conducted after announcement, but before implementation (ie, the 2017 data presented in the current work), of the SDIL found that 70% of UK adults supported the SDIL and 71% thought it would be effective.[34] Finally, a survey of parents of children aged 5–11 years conducted soon after SDIL implementation found that 57% supported its aims.[35] We are not aware of any study exploring change in public acceptability of SSB taxes from before to after implementation.

> **Box 1  Key characteristics of the UK Soft Drinks Industry Levy[37 43 48 49]**
>
> ► Levied on companies importing or manufacturing sugar-sweetened beverages (SSBs), not consumers.
> ► Intentional 2-year delay between announcement (March 2016) and implementation (April 2018) to give manufacturers time to adapt by developing lower sugar products.
> ► Tiered with eligible drinks containing ≥8 g of sugar per 100 mL charged £0.24 (€0.27, US$0.33)/L, those containing ≥5 g but <8 g charged £0.18 (€0.20, US$0.24)/L, and those containing <5 g not charged.
> ► Exemptions for pure fruit juices, milk-based drinks and a number of other smaller categories.
> ► Announcement included a statement that revenue raised would be spent on school sport and school breakfast clubs.
> ► Associated with substantial reformulation of the UK soft drinks market to reduce sugar content.
> ► Associated with complex changes in SSB prices with some categories increasing in price and others decreasing.
> ► Associated with no change in volume of all drinks purchased, but a reduction in sugar purchased from drinks.

Our aim was to determine whether public acceptability, in terms of both support for, and perceived effectiveness of, the SDIL changed between 4 months prior to implementation (ie, 20 months after announcement) and 8 and 20 months after implementation.

## METHODS

We used repeat cross-sectional survey data from the International Food Policy Study (IFPS). We conducted both unadjusted analyses and analyses adjusted for a number of sociodemographic and psychological variables that are potential, or previously reported, correlates of support for, and perceived effectiveness of, the SDIL.

### Sampling, recruitment and data collection

Data were from UK participants in the 2017, 2018 and 2019 waves of IFPS. This is an annual repeat cross-sectional survey conducted in Australia, Canada, Mexico, the UK and the USA with an annual, pragmatic, recruitment target of 4000 adults per country per wave. Data were collected via self-completed web surveys in November–December each year, representing 4–5 months before implementation, but 19–20 months after announcement (2017); 7–8 months after implementation (2018) and 19–20 months after implementation (2019). Respondents were recruited through Nielsen Consumer Insights Global Panel and their partners' panels. Email invitations with unique survey access links were sent to a random sample of panellists within each country after targeting for demographics; panellists known to be ineligible were not invited. Potential respondents were screened for eligibility and quota requirements based on age and sex. UK participation rates (ie, 'the number of respondents who provided a usable response divided by the total number of initial personal invitations requesting participation')[36]

were 7.4%, 11.5% and 4.5% in 2017, 2018 and 2019, respectively.

Respondents provided informed consent prior to completing the survey and received remuneration in accordance with their panel's usual incentive structure (eg, points-based or monetary rewards, or chances to win prizes). A full description of the study methods can be found at wwwfoodpolicystudycom/methods.

### Inclusion criteria

We included in the analysis UK-resident participants in the 2017, 2018 and 2019 IFPS waves who met the following criteria: were aged 18–64 years, provided information on sex at birth and age, and passed data quality checks (provided a valid response to a data quality question; took at least 15 min to complete the survey; and provided a valid response to at least 3 of 20 open-ended measures).

### Variables used in the analysis

The variables used in the analysis, the survey items from which they were derived, response options and how response options were collapsed for analysis are described in table 1.

### Outcome variables

The outcome variables of interest were single-item measures of support for, and perceived effectiveness of, the SDIL collapsed into binary categories of support versus oppose and effective versus not effective as described in table 1.

### Exposure variable

The exposure variable of interest was year—either 2017, 2018 or 2019.

### Potential confounding variables

We adjusted for a number of individual-level sociodemographic and psychological variables in the analysis (see table 1 for details). With the exception of age, education and income sufficiency, these all showed associations with one or both outcome variables in our previous analysis of 2017 data.[34] Given that age and markers of socioeconomic position, such as education and income sufficiency, have been previously, if inconsistently, associated with public support for SSB taxes,[11 13 16 18 24–26 29 30] we included them here, despite no relationship with the outcomes in our previous analysis.

We included single-item measures of attitudes, knowledge and social norms related to sugary drinks; and trust in advice on sugary drinks from health experts and the food and beverage industry. As previous research has indicated that the acceptability of food taxes varies with the stated intentions of these,[9 19 20 27 30 31] we included a preamble to the questions about support for, and perceived effectiveness of, the SDIL outlining the intention of the levy and the stated use of revenue generated.

Sociodemographic variables considered were age in approximately 10-year age bands, sex at birth, whether or not participants had dependent children, and

socioeconomic position. Parental status was included as the SDIL has been particularly framed in terms of potential benefits for children.[37 38] Socioeconomic position was measured using participants' highest educational qualification and perceived income sufficiency. Income sufficiency has previously been associated with financial resources and health outcomes[39] and provides a comparable measure across the range of different economic settings in IFPS.

Current behaviour has previously been associated with perceived acceptability of public health interventions,[33] and we found that SSB consumers were less likely to support the SDIL in 2017. As such, we adjusted for SSB consumption using the Beverage Frequency Questionnaire. This is a 7-day food record that assesses consumption of 17 beverage categories, including caloric and non-caloric beverages.[40] For each beverage category, respondents report the number of drinks and the usual portion size using category-specific images of beverage containers adapted from the Automated Self-Administered 24-hour dietary assessment tool. Participants who reported any consumption of regular fizzy drinks, alcoholic drinks with regular mixers or cocktails that have calories, sweetened fruit drinks, sports drinks or energy drinks over the previous 7 days were considered SSB consumers.

### Analysis

Data were weighted with post-stratification sample weights constructed using population estimates from the UK census based on age group, sex and region. These were used throughout the analysis to reduce the effects of non-response and selection bias. We included 'don't know' and 'refuse to answer' responses as described in table 1, meaning there were no missing data.

Descriptive statistics were used to quantify all variables of interest. Logistic regression models were fitted to explore associations between study wave and the binary measures of support for, and perceived effectiveness of, the SDIL before and after adjustment for other variables. We used separate models to explore support for the SDIL and perceived effectiveness of the SDIL. In these models, support and perceived effectiveness were the outcome variables, study wave was the exposure variable and all other variables in table 1 were covariates. SEs were not clustered. We used the fully adjusted models, and mean values of covariates, to predict the proportion of the population likely to be supportive of the SDIL, and think it would be effective, at each time point.

Data were analysed using Stata V.15.

### Patient and public involvement

Patients and the public were not involved in the design, conduct, analysis or interpretation of the study.

### RESULTS

A total of 25 692 adults took part in IFPS across all included countries in 2017, 28 684 in 2018 and 29 290 in 2019. After

**Table 1** Description of items and response options used in the analysis

| Concept | Item wording (where applicable) | Response options | |
| --- | --- | --- | --- |
| | | All | Categories used in the analysis |
| Age | How old are you? | In years | 18–24 years |
| | | | 25–34 years |
| | | | 35–44 years |
| | | | 45–54 years |
| | | | 55–64 years |
| Sex | What sex were you assigned at birth, meaning on your original birth certificate? | Female | Female |
| | | Male | Male |
| Education | What is the highest level of education you have completed? | Qualifications not listed below, free-text equivalents, Don't know, Refuse to answer | School level |
| | | NVQ Level 4–5, HNC, HND, RSA Higher Diploma, BTEC Higher Level, Degree, Higher Degree, free-text equivalents | Postschool level |
| Income sufficiency | How easy is it to make ends meet? | Neither easy nor difficult, Difficult, Very difficult, Don't know, Refuse to answer | Not easy |
| | | Very easy, Easy | Easy |
| Children | Do you have any children (including stepchildren or adopted children) under the age of 18? | No, Don't know, Refuse to answer | No |
| | | Yes | Yes |
| SSB consumption | (Calculated from Beverage Frequency Questionnaire: reported consumption over last 7 days) | Any consumption of non-diet fizzy drinks, Sweetened fruit juice drinks, Regular sports drinks, Regular energy drinks or Spirits with mixers that have calories | Consumers |
| | | No consumption of above | Non-consumers |
| Social norms | People important to me try not to drink sugary drinks | Neither agree nor disagree, Disagree, Strongly disagree, Don't know, Refuse to answer | Not agree |
| | | Strongly agree, Agree | Agree |
| Attitudes | Sugary drinks taste good | Strongly agree, Agree | Agree |
| | | Neither agree nor disagree, Disagree, Strongly disagree, Don't know, Refuse to answer | Not agree |
| Knowledge | Frequently drinking sugary drinks increases the risk of obesity | False, Don't know, Refuse to answer | Not true |
| | | True | True |
| Expert trust | I trust messages from health experts on sugary drinks | Neither agree nor disagree, Disagree, Strongly disagree, Don't know, Refuse to answer | Not agree |
| | | Strongly agree, Agree | Agree |
| Industry trust | I trust messages from the food and beverage industry on sugary drinks | Neither agree nor disagree, Disagree, Strongly disagree, Don't know, Refuse to answer | Not agree |
| | | Strongly agree, Agree | Agree |

Continued

**Table 1** Continued

| Concept | Item wording (where applicable) | Response options | |
|---|---|---|---|
| | | **All** | **Categories used in the analysis** |
| Support | In 2018 a new sugary drink tax will be/was introduced in the UK. This aims to encourage manufacturers to reduce the sugar in drinks. The money will be spent on breakfast clubs and sports in primary schools. Do you support or oppose this policy? | Strongly support, Support | Support |
| | | Oppose, Strongly oppose, Don't know, Refuse to answer | Oppose |
| Effectiveness | Preamble as above. How effective do you think these kinds of policies would be/are? | Somewhat effective, Mostly effective, Very effective | Effective |
| | | Not at all effective, Don't know, Refuse to answer | Not effective |

BTEC, Business and Technology Education Council; HNC, Higher National Certificate; HND, Higher National Diploma; NVQ, National Vocational Qualification; RSA, Royal Society of Arts; SSB, sugar-sweetened beverage.

removing respondents with missing data on sex at birth and age, and those who did not meet data quality checks, 18 878 (73.5%) respondents remained in 2017, 22 824 (79.6%) in 2017 and 20 968 (71.6%) in 2019. Of these, 4047 were from the UK in 2017, 5549 in 2018 and 4139 in 2019. Among these UK participants, 3104 (76.7%) met the additional inclusion criteria for the current work (ie, aged 18–64 years) in 2017, 4118 (74.2%) in 2018 and 3062 (74.0%) in 2019. Characteristics of the analytical sample (after applying survey weights) are described in table 2.

Table 3 shows the results of logistic regression analyses of associations between survey wave and support for, and perceived effectiveness of, the SDIL, before and after adjusting for the sociodemographic and psychological concepts listed. In unadjusted analyses, both support for, and perceived effectiveness of, the SDIL dropped between 2017 and 2018, but there was little difference in effect estimates in 2018 vs 2017 and in 2019 vs 2017.

In adjusted analyses, there was no evidence that the proportion of participants supporting the SDIL changed between 2017 and either 2018 or 2019. In contrast, the proportion who perceived the SDIL to be effective in 2018 and 2019 was lower than that in 2017. However, the difference in the proportion who perceived the SDIL to be effective was very similar in 2018 vs 2017 and 2019 vs 2017 indicating that the decreased in perceived effectiveness occurred between 2017 and 2018.

Holding all other variables at their mean levels, the adjusted logistic regression model predicted that 70% (95% CI: 68% to 72%) of participants supported the SDIL in 2017, 68% (95% CI: 67% to 70%) in 2018 and 68% (95% CI: 66% to 70%) in 2019 (figure 1). Comparable figures for perceived effectiveness were 72% (95% CI: 70% to 74%) in 2017, 67% (95% CI: 65% to 69%) in 2018 and 67% (95% CI: 64% to 69%) in 2019 (figure 1).

Other variables in the adjusted models were also associated with support for, and perceived effectiveness of, the SDIL (table 3). Greater support for the SDIL was associated with: having a higher level of education, not having children at home, being a non-consumer of SSBs, having social norms to avoid sugary drinks, disliking the taste of sugary drinks, recognising an association between sugary drinks and obesity, trusting health expert messages on sugary drinks and not trusting industry messages on sugary drinks. Individuals aged 35–64 years were also more likely to support the SDIL than those aged 18–24 years, with some evidence of a stepwise increase in likelihood of support across successive age groups. Greater perceived effectiveness of the SDIL was associated with: having children at home, having social norms to avoid sugary drinks, disliking the taste of sugary drinks, trusting health expert messages on sugary drinks and trusting industry messages on sugary drinks. Individuals aged 25–64 years were also less likely to perceive the SDIL to be effective. There was a stepwise decrease in perceived effectiveness across successive age groups to age 54 years, but not thereafter.

## DISCUSSION

### Summary of findings

As far as we are aware, this is the first study to explore whether public acceptability of an SSB tax, operationalised in terms of support and perceived effectiveness, changed from before to after implementation of the tax. It also adds to the small existing literature on public acceptability of implemented (rather than hypothetical) SSB taxes.[28 30 34 35] In this population-based, repeat cross-sectional survey, after adjustment for a range of sociodemographic and psychological covariates, we found that predicted support for the SDIL remained consistently high throughout (68%–70%), with no evidence that support changed from 4 months before to 20 months after implementation. While perceived effectiveness of the SDIL was also high throughout (67%–72%), there was evidence that predicted perceived effectiveness of

**Table 2** Weighted unadjusted characteristics of UK participants; International Food Policy Study, 2017–2019

| Concept | Question wording | Response category | Weighted percentage (95% CIs) | | | |
|---|---|---|---|---|---|---|
| | | | 2017, n=3104 | 2018, n=4118 | 2019, n=3062 | Total, n=10284 |
| Age | How old are you? | 18–24 years | 13 (12 to 15) | 12 (11 to 14) | 11 (10 to 12) | 12 (11 to 13) |
| | | 25–34 years | 22 (20 to 23) | 24 (23 to 26) | 22 (20 to 24) | 23 (22 to 24) |
| | | 35–44 years | 20 (18 to 22) | 21 (20 to 23) | 23 (22 to 25) | 21 (21 to 22) |
| | | 45–54 years | 24 (22 to 26) | 22 (20 to 23) | 20 (19 to 22) | 22 (21 to 23) |
| | | 55–64 years | 21 (19 to 23) | 21 (19 to 22) | 23 (22 to 25) | 22 (21 to 22) |
| Sex | What sex were you assigned at birth, meaning on your original birth certificate? | Female | 48 (46 to 50) | 50 (48 to 52) | 50 (48 to 52) | 49 (48 to 51) |
| Education | What is the highest level of education you have completed? | School level | 61 (59 to 63) | 74 (72 to 75) | 74 (73 to 76) | 70 (69 to 71) |
| Income sufficiency | How easy is it to make ends meet? | Not easy | 61 (59 to 63) | 66 (65 to 68) | 66 (64 to 68) | 65 (64 to 66) |
| Children | Do you have any children (including stepchildren or adopted) under 18? | No | 63 (61 to 65) | 68 (66 to 70) | 65 (62 to 67) | 66 (64 to 67) |
| SSB consumption | Consumed regular fizzy drinks, sweetened fruit drinks, sports drinks, energy drinks in last week | Consumers | 53 (50 to 55) | 44 (42 to 45) | 44 (42 to 47) | 46 (45 to 48) |
| Social norms | People important to me try not to drink sugary drinks | Not agree | 46 (44 to 48) | 52 (50 to 54) | 51 (48 to 53) | 50 (48 to 51) |
| Attitudes | Sugary drinks taste good | Agree | 62 (60 to 64) | 64 (63 to 66) | 59 (57 to 61) | 62 (61 to 63) |
| Knowledge | Frequently drinking sugary drinks increases the risk of obesity | Not true | 10 (9 to 12) | 14 (13 to 16) | 12 (10 to 13) | 12 (12 to 13) |
| Expert trust | I trust messages from health experts on sugary drinks | Not agree | 39 (37 to 41) | 40 (38 to 42) | 41 (39 to 43) | 40 (39 to 41) |
| Industry trust | I trust messages from the food and beverage industry on sugary drinks | Not agree | 73 (71 to 75) | 69 (67 to 70) | 68 (66 to 70) | 70 (69 to 71) |
| Support | In 2018 a new sugary drink tax will be/was introduced in the UK. This aims to encourage manufacturers to reduce the sugar in drinks. The money will be spent on breakfast clubs and sports in primary schools. Do you support or oppose this policy? | Support | 70 (68 to 72) | 66 (64 to 68) | 66 (64 to 68) | 67 (66 to 68) |
| Effectiveness | Preamble as above. How effective do you think these kinds of policies are? | Effective | 71 (69 to 73) | 66 (64 to 67) | 65 (63 to 67) | 67 (66 to 68) |

SSB, sugar-sweetened beverage.

**Table 3** Adjusted ORs (95% CIs) of the association between year and support for, and perceived effectiveness of, the Soft Drinks Industry Levy (SDIL); International Food Policy Study, 2017–2019

| Concept | Question wording (where applicable) | Response category | Support for the SDIL | | Perceived effectiveness of the SDIL | |
|---|---|---|---|---|---|---|
| | | | Unadjusted | Adjusted | Unadjusted | Adjusted |
| Survey wave | Not applicable | 2017 | Reference | Reference | Reference | Reference |
| | | 2018 | **0.84 (0.75 to 0.95)** | 0.93 (0.81 to 1.05) | **0.77 (0.68 to 0.87)** | **0.78 (0.69 to 0.88)** |
| | | 2019 | **0.84 (0.74 to 0.96)** | 0.90 (0.78 to 1.03) | **0.76 (0.67 to 0.87)** | **0.76 (0.66 to 0.86)** |
| Age | How old are you? | 18–24 | | Reference | | Reference |
| | | 25–34 | | 1.07 (0.89 to 1.28) | | **0.75 (0.61 to 0.91)** |
| | | 35–44 | | **1.28 (1.06 to 1.55)** | | **0.50 (0.41 to 0.62)** |
| | | 45–54 | | **1.57 (1.30 to 1.90)** | | **0.46 (0.37 to 0.56)** |
| | | 55–64 | | **1.81 (1.50 to 2.19)** | | **0.47 (0.39 to 0.58)** |
| Sex | What sex were you assigned at birth, meaning on your original birth certificate? | Female | | Reference | | Reference |
| | | Male | | 1.01 (0.92 to 1.13) | | 1.05 (0.95 to 1.16) |
| Education | What is the highest level of education you have completed? | School level | | Reference | | Reference |
| | | Postschool level | | **1.19 (1.07 to 1.32)** | | 1.02 (0.92 to 1.13) |
| Income sufficiency | How easy is it to make ends meet? | Not easy | | Reference | | Reference |
| | | Easy | | 1.07 (0.96 to 1.20) | | 1.02 (0.92 to 1.14) |
| Dependent children | Do you have any children (including stepchildren or adopted children) under 18? | No | | Reference | | Reference |
| | | Yes | | **0.82 (0.73 to 0.92)** | | **1.21 (1.07 to 1.36)** |
| SSB consumption | Consumed regular fizzy drinks, sweetened fruit drinks, sports drinks, energy drinks in last week | Consumers | | Reference | | Reference |
| | | Non-consumers | | **1.12 (1.01 to 1.25)** | | 1.01 (0.91 to 1.12) |
| Social norms | People important to me try not to drink sugary drinks | Not agree | | Reference | | Reference |
| | | Agree | | **1.35 (1.21 to 1.50)** | | **1.35 (1.21 to 1.50)** |
| Attitudes | Sugary drinks taste good | Agree | | Reference | | Reference |
| | | Not agree | | **1.32 (1.21 to 1.50)** | | **1.33 (1.19 to 1.48)** |
| Knowledge | Frequently drinking sugary drinks increases the risk of obesity | Not true | | Reference | | Reference |
| | | True | | **2.78 (2.37 to 3.27)** | | **1.26 (1.07 to 1.48)** |

Continued

**Table 3** Continued

| Concept | Question wording (where applicable) | Response category | Support for the SDIL | | Perceived effectiveness of the SDIL | |
|---|---|---|---|---|---|---|
| | | | Unadjusted | Adjusted | Unadjusted | Adjusted |
| Expert trust | I trust messages from health experts on sugary drinks | Not agree | | Reference | | Reference |
| | | Agree | | **2.36 (2.09 to 2.66)** | | **1.96 (1.75 to 2.19)** |
| Industry trust | I trust messages from the food and beverage industry on sugary drinks | Not agree | | Reference | | Reference |
| | | Agree | | **0.72 (0.63 to 0.82)** | | **1.53 (1.34 to 1.74)** |

Bold indicates statistically significant at the p<0.05 level; adjusted for all concepts listed.
SSB, sugar-sweetened beverage.

the SDIL decreased from 72% before implementation to 67% after implementation. This change was evident 7–8 months after implementation, with no further decrease 12 months later.

### Strengths and weaknesses of methods

Key strengths of the analysis are the large (relatively to other work in the field),[9] population-based, sample; inclusion of a range of sociodemographic, consumption and psychological variables; the context of an implemented, rather than hypothetical, SSB tax in the latter two time points; and consistency of methods across all three time points. Given previous findings showing that support for SSB taxes is greater when revenues are used for health-promoting activities,[9 19 20 27 30 31] we were careful to present the SDIL with revenues ear-marked for health-promotion activities. We also clearly stated that it was an intervention

designed to target manufacturers rather than consumers. Social desirability bias may also be less likely to occur in more anonymous settings such as online surveys.[41]

Participants were recruited using non-probability sampling. Despite the use of weights for age, sex and region, the findings do not necessarily provide nationally representative estimates; and are limited to ages 18–64 years. While the pattern of results in terms of patterns of associations between variables is likely to be generalisable to the UK, the estimates of absolute frequency may not be. Given international differences in dietary public health policy,[42] the pattern of findings may not be generalisable beyond the UK. Although the IFPS takes place in a number of countries, key questions used here were only asked of UK participants. All data were collected after announcement of the SDIL in March 2016.

All variables were self-reported. While all have strong face validity and the Beverage Frequency Questionnaire performs well compared with a 7-day food record,[40] we have not explored validity or reliability of the other measures used. However, many were derived from existing instruments. Further, we were reliant on data availability and did not have consistent information across all three included waves on additional variables that may be of relevance including: household income, age of children in the household and personality traits such as extraversion, agreeableness, conscientiousness, neuroticism or openness to experience. Although adults aged 65 years and older were included in IFPS from 2018 onwards, they were not in 2017.

### Comparison with previous results and interpretation of findings

Overall, there was high support for the SDIL throughout with no evidence of significant change across years. The 68%–70% prevalence of support we found is noticeably higher than the previously reported ranges of 35%–60%[11–27] and a pooled estimate of 42%.[9] We

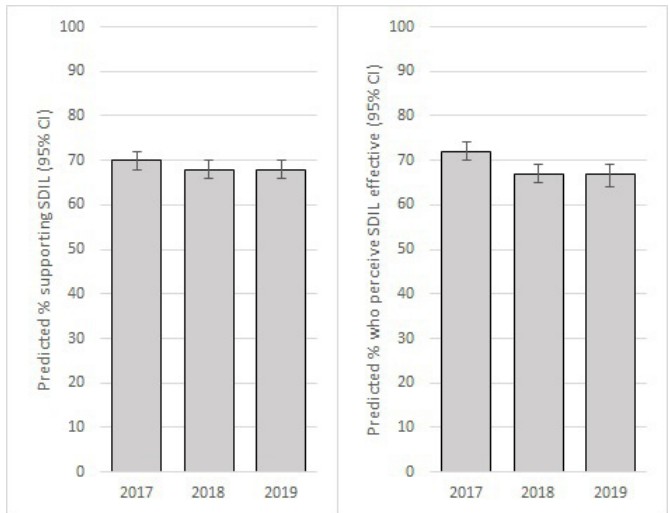

**Figure 1** Predicted percentage (95% CIs) of participants who supported (left) and perceived the Soft Drinks Industry Levy (SDIL) to be effective (right); International Food Policy Study, 2017–2019

propose three potential reasons why support here may be higher than previously reported. First, most previous data on support for SSB taxes have been collected in the context of hypothetical taxes. Previous systematic review evidence suggests that support for governmental public health interventions tends to increase after implementation.[33] While 2017 data were collected 4 months prior to implementation of the SDIL, the policy intention was announced in March 2016, 20 months before 2017 data collection. As such, many participants may have either believed the SDIL had already been implemented, or at least accepted that it was going to be implemented, at the time of 2017 data collection. Comparable pre-announcement data are not available.

Second, other than our previous work using the 2017 data, the only other study of support for the SDIL focused particularly on parents.[35] This found that 1–3 months after implementation, 57% of parents supported what the SDIL was trying to achieve 1–3 months after implementation. We found that those with dependent children were less likely to support the SDIL than those without (overall adjusted predicted support was 70% (95% CI: 69% to 72%) in those without children, and 66% (95% CI: 64% to 68%) in those with children) meaning that our sample including both those with and without dependent children would likely have higher support than one focused exclusively on parents.

Finally, previous research has found that support for SSB taxes consistently increases when it is made clear that revenues will be used for health promotion.[9 19 20 27 30 31] We indicated that the government's stated intention for SDIL revenues was to spend them 'on breakfast clubs and sports in primary schools'. This may have increased support compared with others studies.

While perceived effectiveness also remained high throughout, it decreased from 72% in 2017 to 67% in 2018 and 2019. We propose three potential explanations. First, it is possible that the initial decrease in perceived effectiveness reflects an assumption that the levy achieves its effects via price increases, coupled with limited experience of price increases. The government's stated (and achieved) aim of the SDIL was to prompt reformulation and we were careful to state this in the survey.[43] Despite this, many people may find it difficult to dissociate the concept of 'tax' from price increases and so assume that this is how the levy achieves its effects. This may have been reinforced by temporary signage in stores in spring 2018 explaining that any recent price increases were due to the levy.[44] The true effect of the SDIL on SSB prices was not straightforward with some taxed categories increasing in price and others decreasing.[43] Further, only 44% of parents reported noticing an increase in SSB prices following SDIL implementation.[35] Nevertheless, if participants believed the levy would only work if it increased prices and they did not consistently experience price increases, they could well conclude it was less effective than they would have predicted prior to implementation. Second, the drop in perceived effectiveness between 2017 (pre-implementation) and 2018 (post-implementation)

may reflect the difference between a hypothetical and implemented tax, and that despite similar wording, the measures of perceived effectiveness used in these years were not entirely comparable. Finally, media framing of the SDIL may have changed over time, influencing perceptions of its effectiveness. Although there have been analyses of media coverage of the announcement of the SDIL, we are not aware of any work that has tracked this longitudinally.[45 46]

The associations between sociodemographic and psychological covariates and both support for and perceived effectiveness of the SDIL largely reflect those reported in our previous analysis.[34] As these associations were not the focus of the present work, we refer readers there for a fuller consideration of the interpretation of these associations. In brief, the patterns found largely reflect an intuitive association between more 'public health'-orientated attitudes and beliefs and acceptability of the SDIL.

### Implications of findings

Many structural public health interventions, such as SSB taxes, require government action. This means that political support is an important determinant of implementation of such interventions. Public acceptability may be one important influence on political support. Public acceptability may not just impact on short-term effectiveness via mechanisms such as cross-border shopping, but also on tax longevity and hence long-term effectiveness. Even when written into legislation, such interventions are not necessarily immutable. For example, the SSB tax in Chicago, Illinois was repealed 2 months after implementation,[6] and a tax on high-fat products in Denmark was repealed after a year.[7] In the UK, a proposed tax increase on hot baked goods (the 'pasty tax') was abandoned before implementation following a public outcry.[8] Given this history of repeal of structural interventions, public acceptability is likely to be an important determinant not just of initial implementation but of ongoing longevity and hence long-term impact. That public support for, and perceived effectiveness of, the SDIL remains high even after implementation may help it persist and give confidence to policymakers elsewhere that SSB taxes, and other structural public health interventions, can have high and ongoing public acceptability. This may be particularly important with the recent move in the UK towards more structural policies to address obesity in the last 5 years.[47]

Given our finding of a small drop in perceived effectiveness between before and after implementation of the SDIL, it may be valuable to continue to monitor this. Greater understanding of what makes effective structural public health interventions more and less attractive to the public, and how they can be framed to increased acceptability is also required.

### CONCLUSIONS

We found high levels of support for the SDIL among UK adults and no evidence that this changed between

4 months before implementation and 20 months after. We also found that perceived effectiveness of the SDIL remained high, but there was evidence of a small decrease after implementation in 2018. This may relate to reported complexities in the impact of the SDIL on SSB prices and the difference between reporting on perceived effectiveness of a hypothetical versus implemented policy. While public acceptability of structural public health interventions is recognised as an important determinant of implementation, it may also be an important determinant of policy persistence. Greater understanding of influences on public acceptability of structural public health interventions such as SSB taxes, and how it can be increased, is required.

**Contributors** JA, DP, TLP and MW conceived the idea for this paper. JA analysed the data and drafted the manuscript. JA, TLP, MW, DH and LV read and provided critical comments on the manuscript and approved the final version. DH conceived the idea for the IFPS and secured funding. DH and LV developed the first draft of survey. TLP led the further development of the UK survey instrument, with input from JA, MW, DH and LV.

**Funding** Funding for this project was provided by a Canadian Institutes of Health Research (CIHR) Project Grant (PJT-162167). Additional support was provided by an International Health Grant from the Public Health Agency of Canada (PHAC, no grant number available), and a PHAC-CIHR Chair in Applied Public Health (no grant number available). Collection of 2017 and 2018 UK data was supported by the Health Foundation (no grant numbers available). JA and MW are supported by the Medical Research Council (grant number MC_UU_ 00006/7) and the Centre for Diet and Activity Research (CEDAR) which is a UKCRC Public Health Research Centre of Excellence (grant number MR/K023187/1) funding for CEDAR from the British Heart Foundation, Cancer Research UK, Economic and Social Research Council, Medical Research Council, the National Institute for Health Research, and the Wellcome Trust, under the auspices of the UK Clinical Research Collaboration.

**Disclaimer** Views expressed in this paper are those of the authors and not necessarily those of the above named funders. The study has no affiliations with the food industry.

**Competing interests** None declared.

**Patient and public involvement** Patients and/or the public were not involved in the design, or conduct, or reporting, or dissemination plans of this research.

**Patient consent for publication** Not required.

**Ethics approval** The study received ethical clearance from the University of Waterloo Research Ethics Committee (ORE# 21460 and ORE# 30829). All participants provided informed consent to take part.

**Provenance and peer review** Not commissioned; externally peer reviewed.

**Data availability statement** Data are available upon reasonable request. Data are available directly from the International Food Policy Study team on reasonable request (see www.foodpolicystudy.com).

**ORCID iD**
Jean Adams http://orcid.org/0000-0002-5733-7830

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
