## [Reviewer comments · BMJ Open]

ARTICLE DETAILS

TITLE (PROVISIONAL)	PUBLIC ACCEPTABILITY OF THE UK SOFT DRINKS INDUSTRY LEVY: REPEAT CROSS-SECTIONAL ANALYSIS OF THE INTERNATIONAL FOOD POLICY STUDY (2017-2019)
AUTHORS	Adams, J; Pell, David; Penney, Tarra; Hammond, D; Vanderlee, Lana; White, Martin

VERSION 1 – REVIEW

REVIEWER	Howse, Eloise The University of Sydney, Prevention Research Collaboration, School of Public Health
REVIEW RETURNED	05-May-2021

GENERAL COMMENTS	This is a clearly defined study and research question on an important topic for effective implementation of public health interventions to prevent chronic disease. The paper is well written with a sound study design and analytical methods, and the results well described with implications for policy and practice discussed. Well done to all involved in the study. Some very minor comments and proposed revisions, but I also understand if these are not possible given constraints and limitations in regards to word count and/or number of references allowed by the journal. Introduction page 5 line 25. I would suggest adding in 'perceived' before 'public backlash'. You have acknowledged that public acceptability has no gold standard for measurement. In the cases you cited here there is a possible argument as to the perception of public backlash influencing implementation, and the role the media can play in influencing or framing public acceptability. I'm thinking of work done by Shona Hilton's group re: media representation of the SDIL debate. Eg. Buckton et al 2018 in PLoS One noted the surge in opposition on announcement of the SDIL. lines 28-29. Again it is worthwhile highlighting how media framing of a problem can influence or affect public acceptability with flow on impacts for successful implementation. Eg. Rowbotham et al in SSM https://doi.org/10.1016/j.socscimed.2019.112428. I see you have cited Jeff Niederdeppe's work (ref #13), I think this section could be strengthened by a sentence acknowledging this possible impact.
--

	lines 34-35 - Associations between these variables and support - surprised not to see gender or sex here? If it is included in or controlled for in your analysis. lines 41-48 - The Reynolds et al systematic review https://doi.org/10.1098/rsos.190522 would be useful to cite here given the findings and the focus on perceived effectiveness. page 6 line 27 - I think you measured 3 waves, is that correct? 4 months prior to imp, 8 months after and 20 months after. Just ensuring this sentence matches with the abstract. Methods I assume from Table 1 that you collapsed the indicators of support into 2 categories in order to use logistic regression models. Would be helpful to briefly mention that - under 'Analysis'. It's also not clear from this section whether the analysis adjusted for sex? Discussion Summary of findings - line 8. I would provide the citations for the 'small existing literature' here to make it as easy as possible for the reader to refer to it. Comparisons to previous results and interpretation of findings - page 16, lines 3-20. If the Introduction is revised as per the suggestions earlier re: the role of the media, this section could be strengthened by a sentence or two about the media coverage of the SDIL and possible changes over time which also could have affected the outcomes.
--	--

REVIEWER	Mora Corral, Toni Universitat Internacional de Catalunya, IRAPP
REVIEW RETURNED	30-May-2021

GENERAL COMMENTS	The paper's topic is of interest but I have several severe concerns with this piece of research. I pose my arguments in no particular order.  1. The title is a bit long. 2. Not clear which specific months surveys were addressed, looks like was April 2017 and August 2018 & 2019. In this case, seasonality might have an incidence on self-reported outcomes. 3. Pre-period suffers anticipation effects because the levy was very early announced. 4. Was it available for the public domain where revenues were spent after the tax? 5. Nothing is mentioned about response rates from email invitations. Also, not clear missing patterns. 6. Would consider age non-linear and schooling years for education 7. Most worrying aspect is several measures might be jointly determined by personality traits such outcomes and the rest of self-reported perceptions. The latter would imply to estimate by means of seemingly unrelated equations. 8. Perceived income sufficiency is not a good proxy for income. Do authors dispose total expenditure in Nielsen database? 9. Age of children might condition a different perception. 10. Weights reduce selection bias but not corrected. 11. Were standard errors clustered?
---

	12. Some aspects in concluding section are more likely to be opinions/thoughts rather than mechanisms
--	---

VERSION 1 – AUTHOR RESPONSE

In response to Reviewer 1:

1. page 5, line 25. I would suggest adding in 'perceived' before 'public backlash'. You have acknowledged that public acceptability has no gold standard for measurement. In the cases you cited here there is a possible argument as to the perception of public backlash influencing implementation, and the role the media can play in influencing or framing public acceptability. I'm thinking of work done by Shona Hilton's group re: media representation of the SDIL debate. Eg. Buckton et al 2018 in PLoS One noted the surge in opposition on announcement of the SDIL.

Thank you for pointing out this nuance. We have qualified “public backlash” in this statement with the addition of “perceived”.

2. page 5, lines 28-29. Again it is worthwhile highlighting how media framing of a problem can influence or affect public acceptability with flow on impacts for successful implementation. Eg. Rowbotham et al in SSM <https://doi.org/10.1016/j.socscimed.2019.112428>. I see you have cited Jeff Niederdeppe's work (ref #13), I think this section could be strengthened by a sentence acknowledging this possible impact.

We have added a reference to the Rowbotham paper suggested, and a sentence to capture the importance of media framing: “There is also evidence that framing of SSB taxes, particularly how SSBs are defined and what the stated aims of taxes are, can influence public support.”

3. page 5, lines 34-35 - Associations between these variables and support - surprised not to see gender or sex here? If it is included in or controlled for in your analysis.

Thanks for highlighting this omission. We have noted that the same is true of sex in this sentence.

4. page 5, lines 41-48 - The Reynolds et al systematic review <https://doi.org/10.1098/rsos.190522> would be useful to cite here given the findings and the focus on perceived effectiveness.

Thank you for bringing this highly relevant work to our attention. We have cited it in a number of places and included a sentence capturing the importance of perceived effectiveness for support: “A recent systematic review of randomised controlled trials indicates that providing information on the effectiveness (or ineffectiveness) of government policies leads to significant changes in support for those policies.”

5. page 6, line 27 - I think you measured 3 waves, is that correct? 4 months prior to imp, 8 months after and 20 months after. Just ensuring this sentence matches with the abstract.

Yes, this is correct – there were three measurement points. We have clarified this sentence to reflect all three time points.

6. I assume from Table 1 that you collapsed the indicators of support into 2 categories in order to use logistic regression models. Would be helpful to briefly mention that - under 'Analysis'. It's also not clear from this section whether the analysis adjusted for sex?

We have clarified that the options listed in the final column of Table 1 are the “Categories used in the analysis”, added in the “Outcome variables” section that the measures of support and effectiveness were “collapsed into binary categories of support vs oppose and effective vs not-effective”. We have also clarified in the “Analysis” section that: “Logistic regression models were fitted to explore associations between study wave and the binary measures of support for, and perceived

effectiveness of, the SDIL before and after adjustment for other variables.” [underlining indicates additions]

The analyses are adjusted for sex. This is as described in the section “Potential confounding variables” where it is stated that “Socio-demographic variables considered were age in approximately 10-year age bands, sex at birth...”. However, to further clarify we have indicated in the “Analysis” section that “all other variables in Table 1 were covariates” [Sex is listed in Table 1].

7. Summary of findings - line 8. I would provide the citations for the 'small existing literature' here to make it as easy as possible for the reader to refer to it.

We've added the four relevant references here as suggested.

8. Comparisons to previous results and interpretation of findings - page 16, lines 3-20. If the Introduction is revised as per the suggestions earlier re: the role of the media, this section could be strengthened by a sentence or two about the media coverage of the SDIL and possible changes over time which also could have affected the outcomes.

Thanks for this suggestion. We have added the following sentences as one further potential explanation of the change in perceived effectiveness over time: “Finally, media framing of the SDIL may have changed over time, influencing perceptions of its effectiveness. Although there have been analyses of media coverage of the announcement of the SDIL, we are not aware of any work that has tracked this longitudinally.”

In response to Reviewer 2

1. The title is a bit long.

We have removed 10 words from the title, shortening it to: “Public acceptability of the UK Soft Drinks Industry Levy: repeat cross-sectional analysis of the International Food Policy Study (2017-2019).”

2. Not clear which specific months surveys were addressed, looks like was April 2017 and August 2018 & 2019. In this case, seasonality might have an incidence on self-reported outcomes.

We state in the section entitled “Sampling, recruitment and data collection” that: “Data were collected via self-completed web surveys in November-December each year, representing 4-5 months before implementation, but 19-20 months after announcement (2017); 7-8 months after implementation (2018) and 19-20 months after implementation (2019).” Thus, we do not believe that seasonality is relevant to the interpretation of our findings.

3. Pre-period suffers anticipation effects because the levy was very early announced.

We agree that the context in which the questions about acceptability, and perceived effectiveness, of the SDIL changed over time – in particular that the first data collection point was after announcement but before implementation, whilst the second and third were after implementation. We discuss the implications of this in detail in the final two paragraphs of the “Comparison to previous results and interpretation of findings” section.

We have clarified in the “Article summary” that: “all time points were after announcement of the SDIL in March 2016”. We have also clarified in that introduction that “Our previous population-based survey conducted after announcement, but before implementation (i.e. the 2017 data presented in the current work), of the SDIL found that...” and that “Our aim was to determine whether public acceptability, in terms of both support for, and perceived effectiveness of, the SDIL changed between four months prior to implementation (i.e. 20 months after announcement) and 8 and 20 months after implementation.”

4. Was it available for the public domain where revenues were spent after the tax?

It is stated in Box 1 that “Announcement included a statement that revenue raised would be spent on school sport and school breakfast clubs”. As noted in Table 1, we made clear to participants that “The

money will be spent on breakfast clubs, and sports in primary schools.” Whilst there is publically available information on how much revenue the SDIL raises each year, there is no information on how this is spent.

5. Nothing is mentioned about response rates from email invitations. Also, not clear missing patterns.

We have included information on participation rates in the methods: “UK participation rates (i.e. “the number of respondents who have provided a usable response divided by the total number of initial personal invitations requesting participation”) were 7.4%, 11.5% and 4.5% in 2017, 2018 and 2019 respectively.”

We have clarified in the “Analysis” section that: “We included ‘don’t know’ and ‘refuse to answer’ responses as described in Table 1, meaning there was no missing data.”

6. Would consider age non-linear and schooling years for education

Thank you for the suggestion concerning age. To take account of potential non-linear effects of age, we have chosen to convert it into an ordinal variable in approximately 10-year age bands, as described in the revised version of Table 1. This had led to some minor changes throughout the methods and results section, but no change in the overall interpretation of the findings.

As described in Table 1, we categorise educational qualifications as ‘school level’ or ‘post school level’ and treat it as a binary variable. We do not have information on years of education and hence cannot consider it as a continuous variable – whether linear or not.

7. Most worrying aspect is several measures might be jointly determined by personality traits such outcomes and the rest of self-reported perceptions. The latter would imply to estimate by means of seemingly unrelated equations.

As stated in the introduction section, the key aim of our analysis “was to determine whether....both public support for, and perceived effectiveness of, the SDIL changed between four months prior to...and 20 months after implementation”. Our interest was not in exploring determinants of support for, or perceived effectiveness of, the SDIL where Seemingly Unrelated equations might be helpful. Indeed, we explored associations between the covariates included here and the outcomes of interest in a previous paper.[1] The analysis we have conducted in the current paper explores changes in our outcomes of interest across year, assuming all other variables stayed constant.

As shown in Table 2, even with the use of weights, there are differences in the socio-demographic profiles of participants in different study waves. In particular, the proportion of participants with school level qualifications was substantially higher in 2018 and 2019 (74%) than in 2017 (61%). As such, holding the covariates constant over study years allows us to take into account these changes in the socio-demographic characteristics of participants in different study years, which may in themselves influence the psychological covariates we have included.

We have clarified in the “Analysis” section that: “Despite the use of sample weights, the socio-demographic characteristics of the analytical samples varied across years, most notably in terms of educational attainment (see Table 2) when this did not occur in the population. By holding all covariates constant in the models, we take account of these changes in the socio-demographic characteristics of participants in different study years which may in turn, influence the psychological covariates we have included.”

Unfortunately, we do not have information on personality traits. We have added in the “Strengths and weaknesses of methods” section that: “Further, we were reliant on data availability and did not have consistent information across all three included waves on additional variables that may of relevance including: household income, age of children in the household and personality traits such as extraversion, agreeableness, conscientiousness, neuroticism or openness to experience.”

8. Perceived income sufficiency is not a good proxy for income. Do authors dispose total expenditure in Nielsen database?

Whilst later waves of IFPS do include measures of household income, this was not captured in 2017. Hence, we do not have access to information on household income at all three time points.

As stated in the “Potential confounding variables” section, we use perceived income sufficiency as a proxy for socio-economic position, not income per se. Indeed, income itself is generally included in health-related studies as a proxy of socio-economic position – reflecting the multi-dimensional nature of socio-economic position which includes access to not just financial resources but a range of other intangible resources.[2] It is because of this multi-dimensional nature of socio-economic position that we include both educational qualifications and income sufficiency. We have clarified that: “Subjective measures of income recognise and allow for individual variation in cost of living; financial and other support-in-kind from family, employers and government; wealth; and debts. Perceived income sufficiency is associated with economic resources in a variety of contexts, as well as with a number of health outcomes independent of absolute income.”

As above, we have indicated the absence of information on household income as an additional limitation.

9. Age of children might condition a different perception.

We agree this is possible. However, unfortunately we did not have consistent information on the age of participants’ children (if any). As above, we have indicated the absence of age of children as an additional limitation.

10. Weights reduce selection bias but not corrected.

We agree and this is indicated in the “Analysis” section: “Data were weighted with post-stratification sample weights...to reduce the effects of non-response and selection bias.” We further acknowledge this limitation in the “Strengths and weaknesses of methods” section: “Despite the use of weights for age, sex and region, the findings do not necessarily provide nationally representative estimates”.

11. Were standard errors clustered?

We have clarified that standard errors were not clustered in the “Analysis” section.

12. Some aspects in concluding section are more likely to be opinions/thoughts rather than mechanisms

We agree that it is important to distinguish between findings and interpretation. However, we also believe that some level of interpretation is important to include in the conclusions to help readers understand what our findings may mean. We have edited the “Conclusion” section to make clear which aspects relate to findings from the current study (indicating these with “we found”), and which relate to our interpretation of our findings (indicating these with “this may” or “it may”).

References

1. Pell D, Penney T, Hammond D, Vanderlee L, White M, Adams J. Support for, and perceived effectiveness of, the UK soft drinks industry levy among UK adults: cross-sectional analysis of the International Food Policy Study. *BMJ Open* 2019;**9**:e026698.
2. Galobardes B, Shaw M, Lawlor DA, Lynch JW, Davey Smith G. Indicators of socioeconomic position (part 1). *Journal of Epidemiology and Community Health* 2006;**60**:7.